# The Exact Query Complexity of Yes-No Permutation Mastermind

**Mourad El Ouali [1],* and Volkmar Sauerland [2]** 

[1] Polydisciplinary Faculty Ouarzazate, University Ibn Zohr, Agadir 80000, Morocco

[2] Department of Mathematics, Kiel University, 24118 Kiel, Germany; sauerland@math.uni-kiel.de

* Correspondence: elouali@math.uni-kiel.de

**Abstract:** Mastermind is famous two-player game. The first player (*codemaker*) chooses a secret code which the second player (*codebreaker*) is supposed to crack within a minimum number of code guesses (queries). Therefore, the codemaker's duty is to help the codebreaker by providing a well-defined error measure between the secret code and the guessed code after each query. We consider a variant, called Yes-No AB-Mastermind, where both secret code and queries must be repetition-free and the provided information by the codemaker only indicates if a query contains any correct position at all. For this Mastermind version with $n$ positions and $k \geq n$ colors and $\ell := k + 1 - n$, we prove a lower bound of $\sum_{j=\ell}^{k} \log_2 j$ and an upper bound of $n \log_2 n + k$ on the number of queries necessary to break the secret code. For the important case $k = n$, where both secret code and queries represent permutations, our results imply an exact asymptotic complexity of $\Theta(n \log n)$ queries.

**Keywords:** Mastermind; permutation; query complexity

**MSC:** 91A46

## 1. Introduction

Mastermind is a popular two-player board game invented by Mordecai Meirowitz and released in 1971 [1,2]. Its idea is that a codemaker chooses a secret code of fixed length $n$, where each position is selected from a set of $k$ colors. The second player, codebreaker, has to identify the secret code by a finite sequence of corresponding code guesses (queries), each of which is replied with the number of matching positions and the number of further correct colors. The original game is played by picking pegs of $k = 6$ different colors and placing them into rows with $n = 4$ holes, where the number of rows (allowed queries) for the codebreaker is limited.

Generalizing the situation to arbitrarily many positions and colors, the codemaker selects a vector $y \in [k]^n$ and the codebreaker gives in each iteration a query in form of a vector $x \in [k]^n$. In the original setting, the codemaker's reply is the so called *black-white* error measure, consisting of a pair of numbers, where the first number, black$(x, y)$, is the number of positions in which $x$ and $y$ coincide and the second number, white$(x, y)$, is the number of additional colors which appear in both x and y, but at different positions. In this paper, we consider a variant, called Yes-No AB-Mastermind, which is defined by the following properties

- Both secret code and queries must be repetition-free. This property is indicated by the prefix AB and stems from the AB game, better known as "Bulls and Cows" (cf. [1]), which was even known prior to the commercial version of Mastermind with color repetitions.
- The provided information by the codemaker only answers the question whether or not a query contains any correct position at all. This property is introduced by us and referred to by the term "Yes-No".

**Related Works:** Mastermind and its variants have been analyzed under different aspects. One of the first analyses of the commercial version with $n = 4$ positions and $k = 6$ colors is by Knuth [3] and shows that each code can be cracked in at most 5 queries. Even before the appearance of Mastermind as a commercial game, Erdős and Rényi [4] analyzed the asymptotic query complexity of a similar problem with two colors in 1963 to be $\Theta(n \log n)$. After Knuth's analysis of the commercial game, many different variants of Mastermind with arbitrary code length $n$ and number of colors $k$ have been investigated. For example, Black-Peg Mastermind restricts its error measure between two codes $x$ and $y$ to the single value black$(x, y)$ (i.e., the exact number of positions where both codes coincide). This version was introduced in 1983 by Chvátal [5] for the case $k = n$, who provides a deterministic adaptive strategy using $2n \lceil \log_2 k \rceil + 4n$ queries. Improved upper bounds for this variant and arbitrary $n$ and $k$ where given by Goodrich [6] ($n \lceil \log_2 k \rceil + \lceil (2 - 1/k)n \rceil + k$) and later by Jäger and Peczarski [7] ($n \lceil \log_2 n \rceil + k$ for $k \leq n$ and $n \lceil \log_2 n \rceil + k - n + 1$ for $k > n$) but remained in the order of $\mathcal{O}(n \log_2 n)$. Doerr et al. [8] provided a randomized codebreaker strategy that only needs $\mathcal{O}(n \log \log n)$ queries in expectation and showed that this asymptotic order even holds for up to $n^2 \log \log n$ colors, if both black and white information is allowed. A first upper bound for AB Mastermind was given by Ker-I Ko and Shia-Chung Teng [9] for the case $k = n$, i.e., secret code and queries represent permutations of $[n]$. Their non-constructive strategy yields an upper bound of $2n \log_2 n + 7n$ queries. A constructive strategy by El Ouali and Sauerland [10] reduced this upper bound by a factor of almost 2 and also included the case $k > n$ of *Black-Peg* AB-Mastermind. The term Black-Peg labels the situation that the error measure between secret code and queries is only "black" information, i.e., the number of coinciding positions, while the "white" information (see above) is omitted. El Ouali et al. [11] combined their upper bound of $(n - 3) \lceil \log_2 n \rceil + \frac{5}{2} n - 1$ queries for $k = n$ and $(n - 2) \lceil \log_2 n \rceil + k + 1)$ queries for $k > n$ with a lower bound of $n$ queries, which is implied by a codemaker strategy. It improved the lower bound of $n - \log \log n$ by Berger et al [12]. However, a gap between $\Omega(n)$ and $\mathcal{O}(n \log_2 n)$ remains for this Mastermind variant. Some facts indicate that closing this gap means to improve both bounds. On the one hand, a careful consideration of the partition of the remaining searchspace with respect to all possible codemaker replies might yield a refined codemaker strategy and possibly increase the lower bound. On the other hand, overcoming the sequential learning process of the codebreaker's binary search strategy might decrease the upper bound. The latter presumption is reinforced by the results of Afshani et al. [13], who consider another permutation-based variant of Mastermind. There, the secret code is a combination of a binary string and a permutation, (both of length $n$), queries are binary strings of length $n$, and the error measure returns the number of leading coincidences in the binary string with respect to the order of the permutation. For this setting, which is also a generalization of the popular *leading ones* test problem in black box optimization, the authors prove an exact asymptotic query complexity of $\Theta(n \log n)$ for deterministic strategies but a randomized query complexity of $\Theta(n \log \log n)$.

One of the ultimate goals in the analysis of Mastermind variants is to prove the exact asymptotic query complexity. As mentioned above, closing the asymptotic gap between the lower $\Omega(n)$ bound and the upper $\mathcal{O}(n \log_2 n)$ bound is an unsolved problem for Black-Peg AB Mastermind. A related open question is whether the same asymptotic number of queries is required for both (Black-Peg) Mastermind with color repetition and (Black-Peg) AB Mastermind.

**Our Contribution:** We consider a new variant of AB-Mastermind which is more difficult to play for the codebreaker since the error-measure provided by the codemaker is less informative. Here, for a secret code $y$ the answer info$(\sigma, y)$ to a query $\sigma$ is "yes" if some of its positions coincide with the secret code, otherwise the answer is "no". We first analyze the worst-case performance of query strategies for this Mastermind variant and give a lower bound of $\sum_{j=\ell}^{k} \log_2 j$ queries for $k \geq n$, which becomes $n \log_2 n - n$ in the case $k = n$. The lower bound even holds if the codebreaker is allowed to use repeated colors in his queries. We further present a deterministic polynomial-time algorithm that identifies the secret code. This algorithm is a modification of the constructive strategy of El Ouali et al. [11]. It returns the secret code in at most $(n - 3) \log_2 n + \frac{5}{2} n - 1$ queries in the case $k = n$

and in less than $(n-2)\log_2 n + k + 1$ queries in the case $k > n$. For the important case $k = n$, our results imply the exact asymptotic query complexity of $\Theta(n \log_2 n)$. Since the considered "Yes-No" error measure implies a new variant of AB-Mastermind, there is no previous reference to compare our results to.

## 2. Results

### 2.1. Lower Bound on the Number of Queries

To simulate the worst case, we allow the codemaker to "cheat" in a way that after every query he may decide for a new secret code that is still in agreement with all information he gave so far.

**Theorem 1.** *Let $k, n \in \mathbb{N}$, $k \geq n$ and $\ell := k + 1 - n$. Every strategy for Yes-No AB-Mastermind needs at least $\sum\limits_{j=\ell}^{k} \log_2 j$ queries in the worst case.*

**Proof.** We give a codemaker strategy that implies the lower bound. For $i \in \mathbb{N}$ let $M_i$ denote the set of secrets that are still possible after the $i$-th query has been answered, starting with $M_0 := \{y \in [k]^n \mid \forall i \neq j \in [n] : y_i \neq y_j\}$. Let $M_i^{\text{yes}} \subset M_i$ be the set of secrets that lead to a yes-answer to the $(i+1)$-th query and $M_i^{\text{no}} := M_i \setminus M_i^{\text{yes}}$ the set of secrets that lead to a no-answer. The strategy of the codemaker in round $i+1$ is as follows:

- If $|M_i^{\text{yes}}| \geq |M_i^{\text{no}}|$, pick a secret from $M_i^{\text{yes}}$ (and give the answer yes)
- Otherwise pick a secret from $M_i^{\text{no}}$ (and give the answer no)

By using this strategy, the codemaker achieves for every round $i$ that

$$|M_i| = |M_i^{\text{yes}}| + |M_i^{\text{no}}| \leq 2\max(|M_i^{\text{yes}}|, |M_i^{\text{no}}|) = 2|M_{i+1}|.$$

This implies $|M_i| \geq 2^{-i}|M_0|$. So, for any $i < \log_2(|M_0|)$ we have

$$|M_i| > 2^{-\log_2(|M_0|)}|M_0| = \frac{|M_0|}{|M_0|} = 1,$$

which means that there are still at least two possible secrets left. Since

$$\log_2(|M_0|) = \log_2\left(\prod_{j=\ell}^{k} j\right) = \sum_{j=\ell}^{k} \log_2 j,$$

we obtain the claimed lower bound.  $\square$

**Corollary 1.** *Every strategy for Yes-No Permutation-Mastermind (the case $k = n$) needs at least*

$$\sum_{j=1}^{n} \log_2 j \geq \frac{n \log n - n}{\log 2}$$

*queries in the worst case.*

We also note that the lower bound on the query complexity of Yes-No AB-Mastermind remains of the asymptotic order $n \log n$ if the number of colors is polynomial in the number of positions ($k = P(n)$, $P$ a polynomial).

## 2.2. Upper Bound on the Number of Queries

**Theorem 2.** *Let $k, n \in \mathbb{N}$, $k \geq n$ and $\ell := k + 1 - n$. For $k = n$, there is a strategy for Yes-No AB-Mastermind that identifies every secret code in at most $(n - 3) \log_2 n + \frac{5}{2}n - 1$ queries and for $k > n$, there is a strategy that identifies every secret code in less than $(n - 2) \log_2 n + k + 1$ queries.*

**Corollary 2.** *The exact asymptotic query complexity of Yes-No Permutation-Mastermind is $\Theta(n \log_2 n)$.*

The proof of Theorem 2 resembles the proof of a corresponding result concerning Black-Peg AB-Mastermind [11], except that the information whether a given query contains a *correct but unidentified* position is not derived directly but requires special querying outlined by Algorithm 1 below. In a nutshell (including both cases $k = n$ and $k > n$), the strategy consists of $k$ distinct initial queries, each of which consists of the first $n$ positions of a circularly shifted version of the vector $(1, 2, \ldots, k)$. From the answers of the initial queries, we will be able to learn the secret code position-wise, keeping record about the positions that have already been identified. As long as there are consecutive initial queries $a$ and $b$ with the property that $a$ coincides with the secret code in at least one yet unidentified position but $b$ does not, we can apply a binary search for the next unidentified position in $a$, using $\mathcal{O}(\log_2 n)$ further queries. Such initial queries $a$ and $b$ exist ever after one (usually after zero) but not all positions of the secret code have been identified.

**Proof of Theorem 2.** **The case $k = n$:** We give a constructive strategy that identifies the positions of the secret code $y \in [n]^n$ one-by-one. In order to keep record about identified positions of the secret code we deal with a partial solution vector $x$ that satisfies $x_i \in \{0, y_i\}$ for all $i \in [n]$. We call the non-zero positions of $x$ *fixed* and the zero-positions of $x$ *open*. The fixed positions of $x$ are the identified positions of the secret code. Remember, that for a query $\sigma = (\sigma_1, \ldots, \sigma_n)$ we denote by

$$\mathrm{info}(\sigma, y) := \begin{cases} \text{yes} & \text{if } \{i \in [n] \mid \sigma_i = y_i\} \neq \varnothing \\ \text{no} & \text{otherwise} \end{cases}$$

the information if there is some position in which $\sigma$ coincides with the secret code $y$. For Yes-No AB-Mastermind the related information whether a query $\sigma$ contains a *correct but unidentified* position cannot always be derived directly but must be obtained by guessing one or two modifications of $\sigma$, rearranging those positions that coincide with the partial solution $x$. The required query procedure is summarized as Algorithm 1.

**Example 1.** *Figure 1 illustrates the four distinct cases that are considered by* infoP. *In the first and easiest case (panel (a)) the actual query $\sigma$ does not coincide with the partial solution $x$. Thus, $\sigma$ contains a correct unidentified position if and only if it contains a correct position at all, i.e.,* $\mathrm{infoP}(\sigma, x, y) = \mathrm{info}(\sigma, y)$. *In the second case (panel (b)), $\sigma$ and $x$ coincide in more than one position, namely the positions with colors 3, 9 and 10. The modified query $\rho$ is obtained from $\sigma$ by rearranging these positions in a way that all identified positions get a wrong color while leaving all open positions of $\sigma$ unchanged. This implies that* $\mathrm{infoP}(\sigma, x, y) = \mathrm{info}(\rho, y)$. *Panels (c) and (d) deal with the case that $\sigma$ and $x$ coincide in exactly one position, say $i$. If $x$ already contains a further non-zero position $j$, we obtain $\rho$ from $\sigma$ by swapping positions $i$ and $j$ in $\sigma$ (the positions with colors 3 and 5 in panel (c)). Again, we obtain that* $\mathrm{infoP}(\sigma, x, y) = \mathrm{info}(\rho, y)$. *Finally, if position $i$ is the only yet identified position of the secret code we have to ask two different modified queries to derive* $\mathrm{infoP}(\sigma, x, y)$ *(panel (d)). We obtain the two queries $\rho_1$ and $\rho_2$, each by swapping the identified position (here 3) with another position in $\sigma$, (here with 1 and 2, respectively). While the color of the identified position is wrong in both modifications $\rho_1$ and $\rho_2$, every other position of $\sigma$ coincides with the corresponding position of at least one modification. Therefore,* $\mathrm{infoP}(\sigma, x, y) = \mathrm{no}$ *if and only if* $\mathrm{info}(\rho_1, y) = \mathrm{info}(\rho_2, y) = \mathrm{no}$.

---

**Algorithm 1:** Function infoP

    **input** : Query $\sigma$, partial solution $x$ and secret code $y$
    **output**: Information whether $\sigma$ contains a correct unidentified position

1  **if** $\sigma$ *and* $x$ *do not coincide* **then** answer := info$(\sigma, y)$;
2  **else if** $\sigma$ *and* $x$ *coincide in more than one position* **then**
3     Let $I \subseteq [n]$ be the set of indices where $\sigma$ and $x$ coincide;
4     Let $\pi : I \to I$ be a derangement (a permutation without any fixed position);
5     Obtain guess $\rho$ from $\sigma$ by replacing $\sigma_i$ with $\sigma_{\pi(i)}$ for all $i \in I$;
6     answer := info$(\rho, y)$;
7  **else**
8     Let $i$ be the unique index with $\sigma_i = x_i$;
9     **if** $x$ *has more then one non-zero position* **then**
10         Let $j \neq i$ be another index with $x_j \neq 0$;
11         Obtain guess $\rho$ from $\sigma$ by swapping positions $i$ and $j$;
12         answer := info$(\rho, y)$;
13     **else**
14         Choose $j_1 \neq i$ and $j_2 \neq i$ with $j_1 \neq j_2$;
15         Obtain guess $\rho_1$ from $\sigma$ by swapping positions $i$ and $j_1$;
16         Obtain guess $\rho_2$ from $\sigma$ by swapping positions $i$ and $j_2$;
17         **if** info$(\rho_1, y) =$ info$(\rho_2, y) =$ no **then** answer := no;
18         **else** answer := yes;
19 **return** answer;

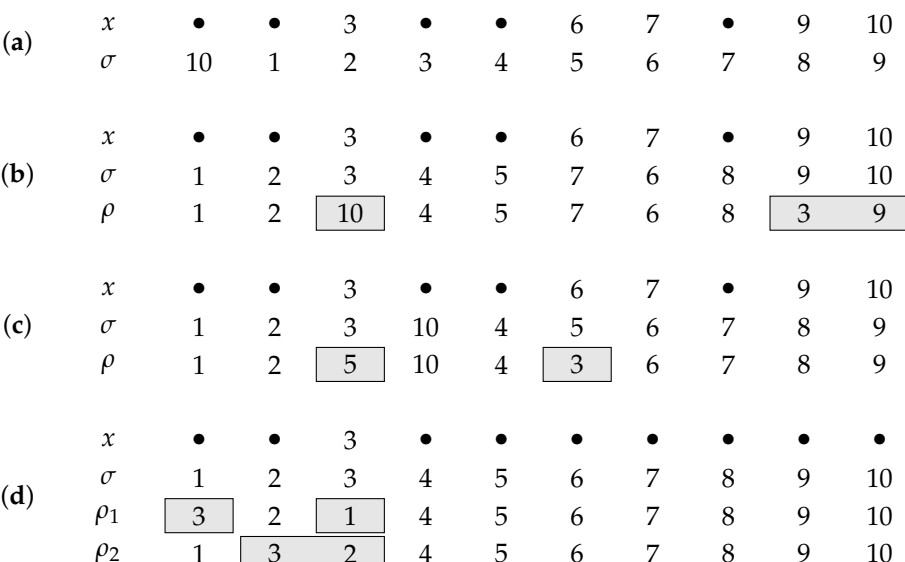

**Figure 1.** Illustrating the cases considered by infoP. Panel (**a**): query $\sigma$ does not coincide with the partial solution $x$; infoP$(\sigma, x, y) =$ info$(\sigma, y)$. Panel (**b**): $\sigma$ and $x$ coincide in more than one position; $\rho$ rearranges these positions of $\sigma$; infoP$(\sigma, x, y) =$ info$(\rho, y)$. Panel (**c**): $\sigma$ and $x$ coincide in exactly one position $i$, but more positions are identified already; $\rho$ is obtained from $\sigma$ by swapping position $i$ with another identified position $j$; infoP$(\sigma, x, y) =$ info$(\rho, y)$. Panel (**d**): Exactly one position is identified and appears to be correct in $\sigma$; two modified queries $\rho_1$ and $\rho_2$ must be defined, each by swapping the identified position with another one; infoP$(\sigma, x, y) =$ no if and only if info$(\rho_1, y) =$ info$(\rho_2, y) =$ no.

    The codebreaker strategy that identifies the secret code $y$ has two phases. In the first phase the codebreaker guesses an initial sequence of $n$ queries that has a predefined structure. In the second

phase, the structure of the initial sequence and the corresponding information by the codemaker enable us to identify correct *positions* $y_i$ of the secret code one after another, each by using a binary search. We denote the vector $x$ restricted to the set $\{s, \ldots, \ell\}$ with $(x_i)_{i=s}^{\ell}$, $s, \ell \in [n]$.

**Phase 1** Consider the $n$ queries, $\sigma^1, \ldots, \sigma^n$, that are defined as follows: $\sigma^1$ represents the identity map and for $j \in [n-1]$, we obtain $\sigma^{j+1}$ from $\sigma^j$ by a circular shift to the right. For example, if $n = 4$, we have $\sigma^1 = (1, 2, 3, 4)$, $\sigma^2 = (4, 1, 2, 3)$, $\sigma^3 = (3, 4, 1, 2)$ and $\sigma^4 = (2, 3, 4, 1)$. The codebreaker guesses $\sigma^1, \ldots, \sigma^n$.

**Phase 2.** Now, the codebreaker identifies the values of $y$ one after another, using a binary search procedure, that we call findNext. The idea is to exploit the information that for $1 \le i, j \le n-1$ we have $\sigma_i^j = \sigma_{i+1}^{j+1}$, $\sigma_i^n = \sigma_{i+1}^1$, $\sigma_n^j = \sigma_1^{j+1}$ and $\sigma_n^n = \sigma_1^1$. findNext is used to identify the second correct position to the last correct position in the main loop of the algorithm.

After the first position of $y$ has been found and fixed in $x$, there exists a $j \in [n]$ such that $\text{infoP}(\sigma^j, x, y) = \text{no}$. As long as we have open positions in $x$, we can either find a $j \in [n-1]$ with $\text{infoP}(\sigma^j, x, y) = \text{yes}$ but $\text{infoP}(\sigma^{j+1}, x, y) = \text{no}$ and set $r := j + 1$, or we have $\text{infoP}(\sigma^n, x, y) = \text{yes}$ but $\text{infoP}(\sigma^1, x, y) = \text{no}$ and set $j := n$ and $r := 1$. We call such an index $j$ an *active* index. Let $j$ be an active index and $r$ its related index. Let $c$ be the color of some position of $y$ that is already identified and fixed in the partial solution $x$. With $\ell_j$ and $\ell_r$ we denote the position of color $c$ in $\sigma^j$ and $\sigma^r$, respectively. The color $c$ serves as a pivot color for identifying a correct position $m$ in $\sigma^j$ that is not fixed, yet. There are two possible modes for the binary search that depend on the fact if $m \le \ell_j$. The mode is indicated by a Boolean variable leftS and determined by lines 5–9 of findNext. Clearly, $m \le \ell_j$ if $\ell_j = n$. Otherwise, the codebreaker guesses

$$\sigma^{j,0} := \left( c, (\sigma_i^j)_{i=1}^{\ell_j - 1}, (\sigma_i^j)_{i=\ell_j+1}^{n} \right) = \left( \sigma_{\ell_j}^j, (\sigma_i^j)_{i=1}^{\ell_j - 1}, (\sigma_i^j)_{i=\ell_j+1}^{n} \right).$$

By the information $\sigma_i^j = \sigma_{i+1}^r$ we obtain that $(\sigma_i^j)_{i=1}^{\ell_j - 1} \equiv (\sigma_i^r)_{i=2}^{\ell_j}$. We further know that every open color has a wrong position in $\sigma^r$. For that reason, $\text{infoP}(\sigma^{j,0}, x, y) = \text{no}$ implies that $m \le \ell_j$. The binary search for the exact value of $m$ is done in the interval $[a, b]$, where $m$ is initialized as $n$ and $[a, b]$ as

$$[a, b] := \begin{cases} [1, \ell_j] & \text{if leftS} \\ [\ell_r, n] & \text{else} \end{cases}$$

(lines 10–15 of findNext). In order to determine if there is an open correct position on the left side of the current center $\ell$ of $[a, b]$ in $\sigma^j$ we can define a case dependent query:

$$\sigma^{j,\ell} := \begin{cases} \left( (\sigma_i^j)_{i=1}^{\ell-1}, c, (\sigma_i^r)_{i=\ell+1}^{\ell_j}, (\sigma_i^j)_{i=\ell_j+1}^{n} \right) & \text{if leftS} \\ \left( (\sigma_i^r)_{i=1}^{\ell_r-1}, (\sigma_i^j)_{i=\ell_r}^{\ell-1}, c, (\sigma_i^r)_{i=\ell+1}^{n} \right) & \text{else} \end{cases}$$

In the first case, the first $\ell - 1$ positions of $\sigma^{j,\ell}$ coincide with those of $\sigma^j$. The remaining positions of $\sigma^{j,\ell}$ cannot coincide with the corresponding positions of the secret code if they have not been fixed, yet. This is because the $\ell$-th position of $\sigma^{j,\ell}$ has the already fixed value $c$, positions $\ell + 1$ to $\ell_j$ coincide with the corresponding positions of $\sigma^r$ which satisfies $\text{infoP}(\sigma^r, x, y) = \text{no}$ and the remaining positions have been checked to be wrong in this case (cf. former definition of leftS in line 5 and line 9, respectively). Thus, there is a correct open position on the left side of $\ell$ in $\sigma^j$, if and only if $\text{infoP}(\sigma^{j,\ell}, x, y) = \text{yes}$. In the second case, the same holds for similar arguments. Now, if there is a correct open position to the left of $\ell$, we update the binary search interval $[a, b]$ by $[a, \ell - 1]$. Otherwise, we update $[a, b]$ by $[\ell, b]$.

---

**Algorithm 2:** Function findNext

---

    **input** :Secret code $y$, partial solution $x \neq 0$ and an active index $j \in [n]$

    **output**:A correct open position in $\sigma^j$

**1** **if** $j = n$ **then** $r := 1$;

**2** **else** $r := j + 1$;

**3** Choose the color $c$ of some non-zero position of $x$;

**4** Let $\ell_j$ and $\ell_r$ be the positions with color $c$ in $\sigma^j$ and $\sigma^r$, respectively;

**5** **if** $\ell_j = n$ **then** leftS := true;

**6** **else**

**7**      $\sigma^{j,0} := \left( c, (\sigma_i^j)_{i=1}^{\ell_j-1}, (\sigma_i^j)_{i=\ell_j+1}^{n} \right) = \left( \sigma_{\ell_j}^j, (\sigma_i^j)_{i=1}^{\ell_j-1}, (\sigma_i^j)_{i=\ell_j+1}^{n} \right)$;

**8**      **if** infoP$(\sigma^{j,0}, x, y)$ **then** leftS := false;

**9**      **else** leftS := true;

**10** **if** leftS **then**

**11**      $a := 1$;

**12**      $b := \ell_j$;

**13** **else**

**14**      $a := \ell_r$;

**15**      $b := n$;

**16** **while** $b > a$ **do**

**17**      $\ell := \lceil \frac{a+b}{2} \rceil$;                           `// position for color c`

**18**      **if** leftS **then** $\sigma^{j,\ell} := \left( (\sigma_i^j)_{i=1}^{\ell-1}, c, (\sigma_i^r)_{i=\ell+1}^{\ell_j}, (\sigma_i^j)_{i=\ell_j+1}^{n} \right)$;

**19**      **else** $\sigma^{j,\ell} := \left( (\sigma_i^r)_{i=1}^{\ell_r-1}, (\sigma_i^j)_{i=\ell_r}^{\ell-1}, c, (\sigma_i^r)_{i=\ell+1}^{n} \right)$;

**20**      **if** infoP$(\sigma^{j,\ell}, x, y)$ **then** $b := \ell - 1$;

**21**      **else** $a := \ell$;

**22** **return** $b$;

---

**Example 2.** *Suppose, that for $n = 10$ the secret code $y$ and the partial solution $x$ are given as in the top panel of Figure 2 and that we have first identified the position with color 1, such that 1 is our pivot color. The initial 10 queries $\sigma^1, \ldots, \sigma^{10}$ together with their current infoP measures are given in the mid panel of Figure 2. We see that the highlighted queries, $\sigma^4$ and $\sigma^5$, can be used for the binary search with findNext, since $\sigma^4$ has a correct not yet identified position but $\sigma^5$ has not. So the active indices are $j = 4$ and $r = 5$ and the corresponding pivot color positions in $\sigma^4$ and $\sigma^5$ are $\ell_j = 4$ and $\ell_r = 5$. The first query of findNext (cf. lower panel of Figure 2) is $\sigma^a$. It begins with the pivot color, followed by the first 3 positions of $\sigma^4$ (positions 2 to 4 of $\sigma^5$) and positions 5 to 10 of $\sigma^4$ (cf. line 7 of findNext). Since infoP$(\sigma^a, x, y) = $ yes, the left most correct but unidentified position in $\sigma^4$ is none of its first 4 positions. Thus, the binary search is continued in the interval $[5, 10]$. It is realized by queries $\sigma^b, \sigma^c,$ and $\sigma^d$, which are composed according to line 20 of findNext (in this case), and finally identifies position 8 with color 5 of the secret code (generally the position left to the left most pivot color position that receives the answer "yes" in the binary search).*

    **The Main Algorithm.** The main algorithm is outlined as Algorithm 3.

    It starts with an empty partial solution and finds the positions of the secret code $y$ one-by-one. The vector $v$ keeps record about which of the initial queries $\sigma^1, \ldots, \sigma^n$ coincides with the secret code $y$ in some open position. Thus, $v$ is initialized by $v_i := \text{info}(\sigma^i, y)$, $i \in [n]$. The main loop always requires an active index. For that reason, if $v_i = $ yes for all $i \in [n]$ in the beginning, we first identify the correct position in $\sigma^1$ (which is unique in this case) by $\lfloor \frac{n}{2} \rfloor + 1$ queries (each swapping two positions of $\sigma^1$) and update $x$ and $v$, correspondingly. After this step, there will always exist an active index.

Every call of findNext in the main loop augments $x$ by a correct solution value. One call of findNext requires at most $1 + \lceil \log_2 n \rceil$ queries if the partial solution $x$ contains more than one non-zero position, and at most $2 + 2\lceil \log_2 n \rceil$ queries (two queries for each call of infoP) if $x$ has exactly one non-zero position. Thus, Algorithm 3 does not need more than $(n - 2)\lceil \log_2 n \rceil + \frac{5}{2}n - 1$ queries to break the secret code inclusive the $n - 1$ initial queries, $\lfloor \frac{n}{2} \rfloor + 1$ queries to find the first correct position, $n - 3$ calls of findNext and 2 final queries.

| (a) | $y$ | 9 | 10 | 6 | 8 | 4 | 2 | 7 | 5 | 1 | 3 | |
| | $x$ | ● | ● | 6 | 8 | ● | 2 | ● | ● | 1 | 3 | |

| | | | | | initial queries | | | | | | | infoP |
|---|---|---|---|---|---|---|---|---|---|---|---|---|
| | $\sigma^1$ | 1 | 2 | 3 | 4 | 5 | 6 | 7 | 8 | 9 | 10 | yes |
| | $\sigma^2$ | 10 | 1 | 2 | 3 | 4 | 5 | 6 | 7 | 8 | 9 | yes |
| | $\sigma^3$ | 9 | 10 | 1 | 2 | 3 | 4 | 5 | 6 | 7 | 8 | yes |
| | $\sigma^4$ | 8 | 9 | 10 | 1 | 2 | 3 | 4 | 5 | 6 | 7 | yes |
| (b) | $\sigma^5$ | 7 | 8 | 9 | 10 | 1 | 2 | 3 | 4 | 5 | 6 | no |
| | $\sigma^6$ | 6 | 7 | 8 | 9 | 10 | 1 | 2 | 3 | 4 | 5 | no |
| | $\sigma^7$ | 5 | 6 | 7 | 8 | 9 | 10 | 1 | 2 | 3 | 4 | no |
| | $\sigma^8$ | 4 | 5 | 6 | 7 | 8 | 9 | 10 | 1 | 2 | 3 | no |
| | $\sigma^9$ | 3 | 4 | 5 | 6 | 7 | 8 | 9 | 10 | 1 | 2 | no |
| | $\sigma^{10}$ | 2 | 3 | 4 | 5 | 6 | 7 | 8 | 9 | 10 | 1 | no |

| | | | | | queries of findNext | | | | | | | infoP |
|---|---|---|---|---|---|---|---|---|---|---|---|---|
| | $\sigma^a$ | 1 | 8 | 9 | 10 | 2 | 3 | 4 | 5 | 6 | 7 | yes |
| (c) | $\sigma^b$ | 7 | 8 | 9 | 10 | 2 | 3 | 1 | 4 | 5 | 6 | no |
| | $\sigma^c$ | 7 | 8 | 9 | 10 | 2 | 3 | 4 | 1 | 5 | 6 | no |
| | $\sigma^d$ | 7 | 8 | 9 | 10 | 2 | 3 | 4 | 5 | 1 | 6 | yes |

**Figure 2.** Panel (**a**): secret code $y$ and partial solution vector $x$. Panel (**b**): the initial queries $\sigma^j$ and their responses infoP$(\sigma^j, x, y)$, indicating if a query and the secret code coincide in any position that has not been identified, yet (i.e., in any 0-position of $x$). Panel (**c**): binary search queries to identify the next secret position. The highlighted subsequences correspond to the subsequences of the initial queries that have been selected to apply the binary search.

---

**Algorithm 3:** Codebreaker Strategy for Permutations

---

1  Initialize $x := (0, 0, \ldots, 0)$;
2  Guess the queries $\sigma^i$, $i \in [n - 1]$;
3  Initialize $v \in \{\text{yes}, \text{no}\}^n$ by $v_i := \text{info}(\sigma^i, y)$, $i \in [n]$;
4  **if** $v_i = \text{yes} \ \forall i \in [n]$ **then**
5  $\quad$ Find position $m$ with a correct color in $\sigma^1$ by at most $\lfloor \frac{n}{2} \rfloor + 1$ queries;
6  $\quad$ $x_m := \sigma^1_m$;
7  $\quad$ $v_1 := \text{no}$;

8  **while** $|\{i \in [n] \mid x_i = 0\}| > 2$ **do**
9  $\quad$ Use $v$ to choose an active index $j \in [n]$; $\qquad\qquad$ // $(v_j = \text{yes}, \ v_{j+1} = \text{no})$
10 $\quad$ $m := \text{findNext}(x, y, j)$;
11 $\quad$ $x_m := \sigma^j_m$;
12 $\quad$ $v_j := \text{infoP}(\sigma^j, x, y)$;

13 Make at most two more queries to find the remaining two unidentified colors;

---

**The case $k > n$:** Let $y = (y_1, \ldots, y_n)$ be the code that must be found. We use the same notations as above.

**Phase 1.** Consider the $k$ queries $\bar{\sigma}^1, \ldots, \bar{\sigma}^k$, where $\bar{\sigma}^1$ represents the identity map on $[k]$ and for $j \in [k-1]$, we obtain $\bar{\sigma}^{j+1}$ from $\bar{\sigma}^j$ by a circular shift to the right. We define $k$ codes $\sigma^1, \ldots, \sigma^k$ by $\sigma^j = (\bar{\sigma}^j_i)^n_{i=1}$, $j \in [k]$. For example, if $k = 5$ and $n = 3$, we have $\sigma^1 = (1, 2, 3)$, $\sigma^2 = (5, 1, 2)$, $\sigma^3 = (4, 5, 1)$, $\sigma^4 = (3, 4, 5)$ and $\sigma^5 = (2, 3, 4)$. Within those $k$ codes, every color appears exactly once at every position and, thus, there are at least $k - n$ initial queries that do not contain any correct position. Since $k > n$, this implies

**Lemma 1.** *There is a $j \in [k]$ with* info$(\sigma^j, y) = $ no.

**Phase 2.** Having more colors than positions, we can perform our binary search for a next correct position without using a pivot color. The corresponding simplified version of findNext is outlined as Algorithm 4.

---

**Algorithm 4:** Function findNext for $k > n$

    **input** : Code $y$, partial solution $x \neq 0$ and an active index $j \in [k]$
    **output**: A position $m$ that is correct in $\sigma^j$

1  **if** $j = n$ **then** $r := 1$;
2  **else** $r := j + 1$;
3  $a := 1, b := n$;
4  **while** $b > a$ **do**
5     $\ell := \lceil \frac{a+b}{2} \rceil$;                          `// mid position of current interval`
6     Guess $\sigma := \left( (\sigma^r_i)^{\ell-1}_{i=1}, (\sigma^j_i)^n_{i=\ell} \right)$;
7     $s := $ infoP$(\sigma, x, y)$;
8     **if** $s = $ yes **then** $a := \ell$;
9     **else** $b := \ell - 1$;
10 **return** $a$;

---

Using that version of findNext also allows to simplify our main algorithm (Algorithm 3) by adapting lines 2 and 3, and, due to Lemma 1, skipping lines 4–7, as findNext can be already applied to find the first correct position. Thus, for the required number of queries to break the secret code we have: the initial $k - 1$ queries, a call of the modified findNext for every but the last two positions and one or two final queries. This yields that the modified Mastermind Algorithm breaks the secret code in at most $(n-1) \lceil \log_2 n \rceil + k + 1$ queries. $\quad \square$

## 3. Conclusions

We showed that deterministic algorithms for the identification of a secret code in Black-Peg AB-Mastermind can be modified and applied to Yes-No AB-Mastermind. The latter is a new variant of AB-Mastermind which is harder to play for the codebreaker since a less informative error measure is provided. The Yes-No measure only returns the information whether a query and the secret code coincide in any position, while the Black-Peg measure is the number of positions in which both codes coincide. Nevertheless, we proved that the best known asymptotic upper bound for Black-Peg AB-Mastermind does also apply to Yes-No AB-Mastermind, by adapting the corresponding constructive querying strategy. Utilizing a simple codemaker strategy, we further derived corresponding lower bounds for Yes-No AB-Mastermind. Another challenge with AB-Mastermind is that no color repetition is allowed in a query whereas most strategies for other Mastermind variants exploit the property of color repetition. While for most Mastermind variants there is a gap between lower and upper bounds on the worst case number of queries to break the

secret code, our results imply that this number is $\Theta(n \log n)$ for the most popular case $k = n$ of Yes-No AB-Mastermind, which is also referred to as Yes-No Permutation-Mastermind. The same is true for the case $k = c \cdot n$ with constant $c$. To our knowledge, this result is a first exact asymptotic query complexity proof for a multicolor Mastermind variant, where both secret code and queries are chosen from the same set, here $[k]^n$.

A future challenge will be studying the static variant of Yes-No AB-Mastermind (where the codebreaker must give all but one queries in advance of codemaker's answers). Lower and upper bounds for static Black-Peg AB-Mastermind were provided as $\Omega(n \log n)$ and $\mathcal{O}(n^{1.525})$, respectively [14].

**Codeavailability:** We provide Matlab/Octave implementations of the codebreaker strategy via GitHub, a permanent version of which is archived in a public zenodo repository [15].

**Author Contributions:** Conceptualization, M.E.O. and V.S.; methodology, M.E.O.; software, V.S.; validation, M.E.O. and V.S; writing—original draft preparation, M.E.O. and V.S.; writing—review and editing, M.E.O. and V.S.; visualization, V.S.; All authors have read and agreed to the published version of the manuscript.

**Funding:** This research received financial support by DFG within the funding programme Open Access Publizieren.

**Conflicts of Interest:** The authors declare no conflict of interest.

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
