# Peer review of "The Exact Query Complexity of Yes-No Permutation Mastermind"

_games, doi:10.3390/g11020019_

Round 1

Reviewer 1 Report

The reference section is repeated and both start in #13.

 Most late articles on Mastermind or Black Box optimization problems are not referenced. There are many papers that use evolutionary algorithms to solve mastermind, for instance, and compare actual results to theoretical bounds.

Implementation should be given in an open source environment, not a proprietary one. If it uses M-code (the language that's implemente in Matlab) it should work in Octave too.

Reviewer 2 Report

[The overall impression]

This article is an extending work of reference [21] and focus on binary version of AB-Mastermind game where the codes and the guesses are permutations. The authors give a lower bound and upper bound on the query complexity of this game and show that the bounds are tight. 

Despite the answers of the queries from codemakers are restricted to be yes or no only, the query complexity stays in the order of O(n\log n). The results basically investigate the Yes-No AB-Mastermind games of the two conditions: either there are the same number of positions and colors or the game has more colors than the positions though they are repetition-free. 

[Evaluation] 

The contribution might be not very significant because of the game instance is restricted to permutations and guessing the codewords and specify the positions by binary search are intuitive.

The main theorem and the related proofs are almost placed in the appendix yet actually the readers require more effort to comprehend the results since the description and explanation in Sect. 2.2 are not clear enough. 

Besides, the paper is not written in a reader-friendly way. Names and definitions of the problems are not consistent and confusing. For instance, at line 30, "... In this paper, we deal with the
combination of Yes-No Mastermind and the AB game." is not clear to me because the AB game is not clearly defined.  

I suggest the authors to give an example to illustrate the main algorithm as well as the functions in the two phases. 

[Minor comments]

Line 84: S_n needs to be well-defined. 

Line 92-93: "... similar to our strategy for Black-Peg AB-Mastermind presented in [21]." This sentence reveals who the authors are and should be avoided. 

Line 206--: The bibliography is redundant. 

Reviewer 3 Report

In this paper, the authors analyzed a variant of the famous Mastermind game and showed that a deterministic algorithm may be applied to tackle this game (and verified its complexity). The paper is well-written and clear.

Round 2

Reviewer 1 Report

An extensive revision of English has not been made. "popular two players zero sum game" should be rephrased, and that's just the first sentence; most sentences, although formally comprehensible, do not sound right. There are also typos that have not been corrected from the first version: "Therefor" in the abstract. The authors should be more careful when submitting a manuscript, and at least spell check it before submitting the final version. Not even the word "MasterMind" is written uniformly across the whole paper. Also, sentences like this in the conclusions "One challenge of this variant
is that codemaker answers queries only with the information whether any position is correct." either contradict the introduction, or actually want to say "whether there's any correct position". That's something totally different.

While Mastermind was "invented" in the seventies (not 1970, please check your sources), it's based on older games like "cows and bulls". It's not relevant to the paper, but you should at least check the Wikipedia page on the game your paper deals with (and reference). Besides, you  talk about the AB game, saying it's "anterior", but you don't provide a reference for this either.

The "Related works" section should include the bounds reached, and in which context. Also, and in general, no paper provides strategies for selecting queries. Queries may still need exponential time to be found.

Besides, there should be a justification of this new variant of the AB-Mastermind, and how its results should extend to the normal mastermind. Its main difference with respect to black-peg mastermind is that in the latter case we also say how many pegs are correct; it's not clear how much more difficult that could get. Additionally, and later on in the paper, you mention that the codemaker could cheat, so that's another variant of the game (which has got its own literature), like this paper by Goodrich https://ieeexplore.ieee.org/abstract/document/6239593

Also, is the formula sum(j=l,k) log 2 j correct? That's still n log 2 n, unless you want to provide a tighter bound.

Reviewer 2 Report

The revision is greatly improved but there are still minor revisions required. 

Line 120: "... positions of shifted versions of the vector (j)_{j\in [k]}" -> I don't see it is guaranteed to be a permutation as it is not repetition-free for certain. 

Line 4 of Algorithm 1: ... without any fixed position);

Page 7: The definition of \sigma^{j,0} is not clear as it should be like a permutation. I don't see how color c is placed in the permutation.

Line 2 in the caption of Figure 2: infoP(\sigma^j, y, x) -> infoP(\sigma^j, x, y).

Line 8 of Algorithm 4: if s is 'yes'  (because the return value of infoP(\sigma, x, y) is 'yes' or 'no'.

Make sure all the index 'l's are consistent. Some of them are 'l's but some of them are '\ell's.
